# Transparent Films Thickness Mapping Highlighting the Viscosity Effect of Elastic Layers Made by Sol–Gel Process with an In-House Ellipsometer

Océane Guillot, Amira Guediche, Mathieu Lafarie, Amandine Moiny, Théo Brockhouse and Hervé Piombini *

CEA, DAM Le Ripault, F-37260 Monts, France
* Correspondence: herve.piombini@cea.fr; Tel.: +33-247344074

**Abstract:** New optical coatings are currently developed to mitigate the shockwave generated by nanosecond lasers in high-power laser systems such as the MegaJoule laser (LMJ). These shockwaves seem responsible for the damage growth observed on optical components. A possible solution for shockwave mitigation is using ormosil (organically modified silicate) coatings made by the sol–gel method with thicknesses of a few microns. Unfortunately, the sol–gel solution exhibits a viscous behavior, and thus, the deposited layers are heterogeneous in thickness. An experimental ellipsometer has been designed to measure this heterogeneity and highlight the viscoelastic properties of the layers responsible for self-healing effects that were observed when these layers were scratched. This ellipsometer allows us to know the refractive index of the coating and therefore its density. Density and thickness are the two essential parameters for determining the speed of sound and the modulus of elasticity of the layer, which indicate the ability of the layer to attenuate more or less elastic waves or shock waves.

**Keywords:** ellipsometer; ormosil; PDMS; sol–gel; thickness homogeneity; transparent thick film; viscoelasticity





## 1. Introduction

The Military Applications Division (DAM) of the French Atomic Energy and Alternative Energies Commission (CEA) is building the Megajoule laser (LMJ) [1,2], whose maintenance costs due to the replacement of optical components damaged by LMJ beams may be high [3] because the optical components must operate close to optical damage thresholds [4]. These optical components are made of fused silica and each face is coated with an antireflective layer. Furthermore, the output energy of this high-power laser system is limited in practice, primarily by laser damage occurring in the ultra-violet range. At $3\omega$ (351 nm), laser damage mainly occurs on the output side of the components or at least is more severe than on the front surface [5]. Demos et al. (2013) described the various successive processes observing the output face with time-resolved microscopy [6]. The production of shockwaves observed by shadowgraphy [7] during the laser-matter interaction with the optical component is responsible for structural changes inside the fused silica, detected by Raman spectroscopy [5] or by X-ray microtomography [8] and simulated by Kubota et al. (2001) [9]. These shockwaves produce mechanical damage mechanisms such as spallation [4,6,10,11].

Some materials with specific properties (high porosity, viscoelasticity) can mitigate shockwaves and elastic waves. Rayate et al. (2017) used viscous materials (PDMS, Teflon. . .) to dampen vibrations in the boring bar of a machine tool and so to improve the surface roughness [12]. Lee et al. (2020), using a laser-induced shockwave technique, probed the ability of the dynamic bond exchange to dissipate shockwave energy in a series of well-defined polydimethylsiloxane (PDMS) with a variable cross-link density [13]. According to Kazemi-Kamyab et al. (2011), porous materials were known to attenuate the

propagation of shock waves [14]. Polysiloxanes such as polydimethylsiloxanes (PDMS) are interesting polymers to integrate into an optical coating. To protect the optics from laser damage and increase their lifetime, Compoint (2015) inserted a PDMS elastic layer on the rear face of the current LMJ optics between the silica substrate and the antireflective film to attenuate the shockwaves [15]. Wolf et al. (2018) indicated that PDMS is optically transparent between 240 and 1100 nm and has good heat resistance (thermal degradation at temperatures > 400 °C) [16]. Kopetz et al. (2007) measured the refractive index of PDMS as a function of temperature. Its refractive index is 1.41 ($\lambda$ = 589 nm) at room temperature [17]– near silica: n = 1.46 ($\lambda$ = 589 nm). Wu et al. (2002) carried out the absorption spectrum of a 10 mm thick sample of PDMS. The spectrum showed that the sample absorbs less than 5% of incident UV illumination for the wavelength ranging from 365 to 436 nm [18]. Besides its optical properties and the stability of its physical properties (viscosity, dielectric properties) with temperature, PDMS has some degree of deformability thanks to its chain mobility (pronounced flexibility of the -[Si–O]x–chain segments) [19,20] that gives to the resulted ormosils viscoelastic properties. Thus, PDMS seems to be an excellent polymer candidate to combine with silica to obtain ormosils with high transmission in the ultraviolet (UV), visible (Vis), and near-infrared (NIR) and damping properties. Antireflective coatings using tetraethyl orthosilicate (TEOS) as a precursor and hydroxyl-terminal PDMS as a modifier were prepared by Zhang et al. (2010, 2012) [21,22]. Transparent elastic thin films are being developed [22] and could be used for this purpose. They are made of silica and polydimethylsiloxane (PDMS), these mixes are organically modified silicates [23] called PDMS-based ormosils or Type C Ormosils [24]. This family of PDMS-based ormosil is obtained through a sol–gel process [23] and deposited as films by spin coating on fused silica substrates. PDMS is used as a tunable parameter to obtain a range of films [25–27] or bulk materials [28] with variable elastic moduli. In addition, Compoint (2015) showed in his thesis that the ormosil layers reveal self-healing properties [15], and Guediche et al. (2021) proved that the self-healing properties of these films are due to their viscoelasticity properties [27]. Experiments demonstrated that a scratch on ormosil layers with 40% in mass of PDMS can disappear after some time [15,27]. Usually, the thickness of antireflective films used on the LMJ optical components is approximately 70 nm (for 3 $\omega$ lasers) according to Avice et al. (2017) [29]. To mitigate shockwaves, the thickness of ormosil films should reach at least a few microns.

Compared to the antireflective films, these thick films are challenging to deposit by spin coating mostly because of the viscosity of the solutions we use [15]. One of the defects of spin-cast films is the difference in thickness between the center and the edge of its surface: Huang and Chou (2003) consistently got thicker films in the center and thinner films on the edge regardless of the viscosity of the solution [30]. Consequently, each layer reveals a heterogeneous thickness. This heterogeneity could be measured by mapping with the reflectometer we developed a few years ago but could not be measured with our spectrophotometers [31,32]. If this thickness difference is insignificant for spin-casted films of 70 nm (a few nanometers), the differences are a lot more significant when films are thicker. Our study samples are smaller (50 mm diameter or 50 mm$^2$) than the optical components installed on the LMJ (400 mm × 400 mm square). The deposit method leads to a 150 nm peak-to-valley gap for a 2 $\mu$m thick layer. The viscosity of the primary solution of PDMS-based ormosil also causes this high value of thickness variation. These local thickness differences lead to modifications of the transmitted wave surface or local micro-slopes [33]. Thus, we would like to monitor the stability of the layer thickness over time by studying the influence of the layer viscosity on the thickness variation over time. For this purpose, an experimental ellipsometer at 633 nm has been developed and is presented in this paper. This experimental setup was able to determine thickness variations of transparent thin films through maps from reflection measurements with precision under 10 nm. It also had the advantage of being well-adapted to transparent substrates (eliminating rear reflections by a diaphragm). This ellipsometer allows us to know the refractive index of the coating and therefore its density. Density and thickness are the two essential parameters for determining

the speed of sound and the modulus of elasticity of the layer, which indicate the ability of the layer to more or less attenuate elastic waves or shock waves.

In a previous study [15,27], we showed that PDMS-based ormosils have self-healing properties when the wt.% of PDMS reaches 40, as depicted in Figure 1. After 14 days, the scratches (Figure 1c) were significantly smaller relative to the initial image (Figure 1a) when observed through optical microscopy. The kinetics of self-healing has two time constants, a short time and a long time [27].

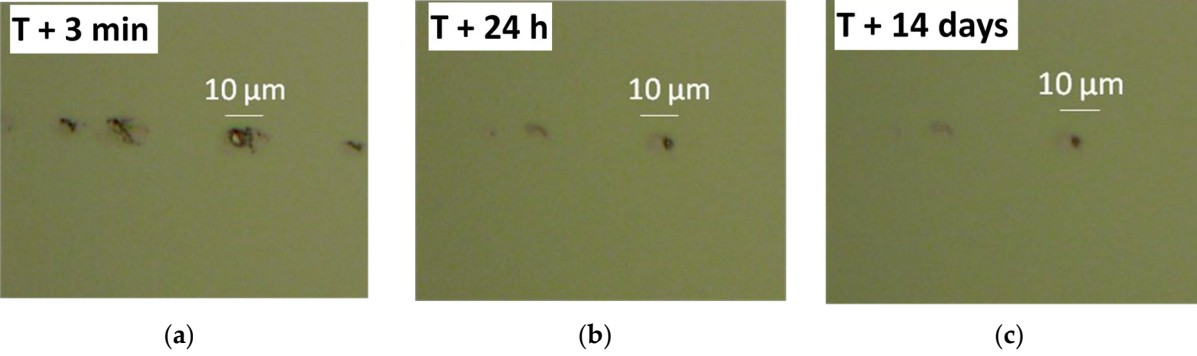

**Figure 1.** Indented layer evolution of a 40 wt.% PDMS-based ormosil at t = 3 min (**a**), t = 24 h (**b**), and at t = 14 days (**c**).

Two hypotheses were made to explain this mechanism:

- a swift mechanical response which allows the defect to be significantly reduced thanks to the elastic properties of the layer, then a long-term chemical response corresponding to the reformation of chemical bonds that sealed the scratch;
- a mechanical response that takes into account the viscoelastic properties of the layer.

The first hypothesis was verified by PMIRRAS (Phase Modulation InfraRed Reflection Absorption Spectroscopy) experiments. For this purpose, scratched layer evolution over time was followed at the same position, at ambient temperature in a dedicated cell under a nitrogen atmosphere. No change in the spectral response was detected, even after several hours. For the second hypothesis, the self-healing process was explained by a time-dependent viscoelastic model. This model corresponds to an association between Voigt-Kelvin's model and a Maxwell model called the Burgers model. The same strain was taken for the two associated models. The agreements between measurements and calculations indicate that the self-healing phenomenon is linked to the viscosity of the layer [27].

In this paper, the heterogeneity problem of thick layers is introduced, followed by the setup of a homemade ellipsometer. Our ellipsometer is based on the angular scanning of the reflected power of s- or p-polarized light and the determination of the Brewster angles as Tikhonov and Lyamets (2019) [34] determined the birefringent indexes. We extend this method to the determination of the layer thicknesses. After validation of the design through the study of two well-known materials, fused silica and BK7 (borosilicate crown glass), the results are also compared with an RC2 ellipsometer from J. A. WOOLLAM Company [35,36] that only works with the sample mounted horizontally. Finally, we reveal that the thick layers made of 40 wt.% PDMS-based ormosil are not stable over time because of their viscosity, they evolve.

## 2. Materials and Methods

### 2.1. Phase Variation in Transmission According to the Thickness Homogeneity

Based on the specifications of the M1 mirrors given by Grosset-Grange et al. (2007) (±0.1 μrad in wavefront tilt stability and 0.3 μrad in root mean square (RMS) of wavefront local slopes for 1 h [37]) and the fact that these slopes are conservative along the optical path, the thickness heterogeneity can be estimated. Thickness variations depend on the deposi-

tion process and are generally homothetic to the deposited thickness t. They correspond to a variation of a wavelength often called centering wavelength $\lambda_c$ and defined by:

$$\lambda_c = 4 \cdot n \cdot t \cdot \cos \theta_L, \tag{1}$$

$$t = \frac{\lambda_c}{4 \cdot n \cdot \cos \theta_L}, \tag{2}$$

with n the refractive index and $\theta_L$ the angle of refraction of the beam in the layer. From it, one can easily calculate the variation of the phase of a layer induced by heterogeneity [33] and, consequently, its gradient

$$\frac{d\varnothing_T}{dx} = a \frac{d\varnothing_T}{d\lambda} \tag{3}$$

For a 2000 nm layer of colloidal silica, the gradient is about ten times greater than the gradient of a 200 nm layer.

In the case of a 200 nm layer (green curve in Figure 2), with a local over-thickness of 10% (depicted by the pink curve in Figure 2), the phase shift is about 4.6° or 80 mrad at 358 nm. The phase shift for a 2000 nm layer is about ten times faster, resulting in slope variations at the output of the optical component that are ten times more important at constant heterogeneity. Thus, it is crucial to have thick sol–gel layers with homogeneous thickness to keep a flat front wave with weak distortions. Consequently, an accurate characterization of the spatial homogeneity of layer thicknesses is required.

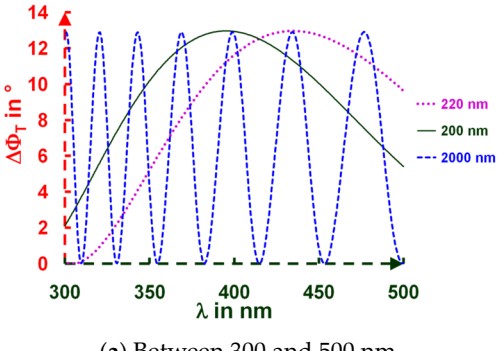

(**a**) Between 300 and 500 nm.

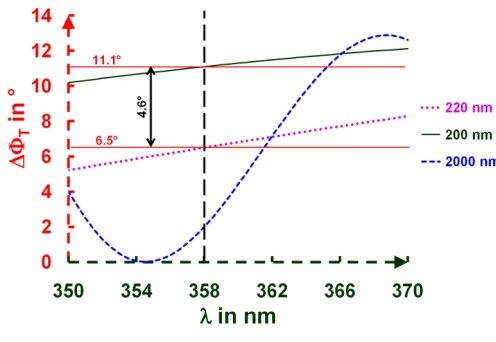

(**b**) Between 350 and 370 nm

**Figure 2.** (**a**) Variation of the phase transmission as a function of wavelength for three-layer thicknesses (200, 220–corresponding to a 10% variation of the 200 nm layer–and 2000 nm) of colloidal silica with a refractive index of 1.22 and an extinction coefficient k of $10^{-4}$ on a substrate having a refractive index of 1.5. (**b**) Zoom of Figure 2a between 350 and 370 nm.

### 2.2. Principle of Method of Characterization

The principle of this experimental setup is based on the transparent thin-film theory given by Abelès (1950) [38] or Macleod (2018) [39]. For a layer deposited on a substrate, the reflection intensity according to the polarization s or p is a function of the film thickness and the angle of incidence θ.

For a layer of thickness t, of refractive index $\eta_L$, and an oblique incident beam, the reflection is given by:

$$R = \rho \cdot \rho* = \left( \frac{\eta_0 - Y}{\eta_0 + Y} \right) \cdot \left( \frac{\eta_0 - Y}{\eta_0 + Y} \right)^* \tag{4}$$

where $\eta_0$ and Y are, respectively, the optical admittances of the incidence medium (air) and the stack (substrate S + layer L).

Y is obtained by the assembly matrix of the stack.

$$\begin{bmatrix} B \\ C \end{bmatrix} = \begin{bmatrix} \cos \delta & i \cdot \frac{\sin \delta}{\eta_L} \\ i \cdot \eta_L \cdot \sin \delta & \cos \delta \end{bmatrix} \begin{bmatrix} 1 \\ \eta_S \end{bmatrix} \text{ with } \delta = \frac{2 \cdot \pi \cdot n_L \cdot t \cdot \cos \theta_L}{\lambda} \text{ and } Y = \frac{C}{B}, \tag{5}$$

$\eta_L$ is defined as the tilted optical admittance and is given by:

$$\eta_L = n_L \cdot \cos \theta_L \quad \text{for s polarization,} \tag{6}$$

$$\eta_L = \frac{n_L}{\cos \theta_L} \quad \text{for p polarization,} \tag{7}$$

Snell's law links the angles of incidence $\theta$ and refraction $\theta_L$ between the air and the film:

$$n_0 \cdot \sin \theta = \sin \theta = n_L \cdot \sin \theta_L \tag{8}$$

So, thickness variations imply variations in the reflection for a given incidence and polarization.

*2.3. Experimental Setup*

An ellipsometer is an excellent device for making measurements on an opaque layer or substrate (silicon wafer) [40,41], but there are technical challenges to overcome for measurements on a transparent layer [42]:

- the rear face reflection;
- detector noises;
- temporal fluctuations of the light source.

The development of the experimental ellipsometer took less than one year as we already had the experience and the right equipment [31,43–46]. It was thus possible to design it to overcome some drawbacks. The backside reflection from the rear side of the substrate was reduced or suppressed by using a diaphragm. Noises of sensors were measured, and temporal fluctuations of the laser source were recorded in real-time and considered in each measurement. A holder was designed to hold the sample in a vertical position, as the LMJ components. On our WOOLLAM ellipsometer, it is impossible to hold the sample vertically.

Figure 3 schematically describes the experimental setup, and a picture is displayed in Figure 4. The ellipsometer could be used in either reflection or transmission, but only the reflection part is used for this application. The laser source was a 6 mW continuous-wave Helium-Neon (HeNe) laser (CWL) rectilinearly polarized; emitting at 633 nm so the ellipsometer was monochromatic. Its polarization was tuned at 45° from the axis of a Glan-Thompson prism. This prism was placed on a motorized rotating stage to choose either s-polarization or p-polarization. Previously, the axis orientation of the prism was determined based on Malus' law as we did in the past [46]. If we note the angle $\varphi$ between the direction of the laser polarization and the plane of the polarizer, the Malus law relates the intensity of light I passing through the polarizer [46] to the initial intensity $I_0$.

$$I = I_0 \cdot \cos^2(\varphi - \varphi_0) \tag{9}$$

where $\varphi_0$ is the polarization angle of the HeNe laser. Once this angle $\varphi_0$ was determined by looking at the beam extinction, we shifted this polarizer angle to $\pm 45°$ to get either p-polarized only or s-polarized only.

The front surface of the chopper SR540 by Stanford Research (Sunnyvale, CA, USA) was metalized to reflect a portion of the laser beam for reference measurement. Since the reflection was very diffuse, we added a 50 mm diameter condenser to focus the beam into an integrating sphere to be collected by a P1 photodiode. The photodetectors used were HUV 1100, developed by Perkin Elmer Optoelectronics (Fremont, CA, USA). They were connected to lock-in amplifiers (SR830 from Stanford Research) coupled with a chopper to modulate the signals at a specific frequency. The part of the incident beam transmitted by the chopper went onto the sample, and the reflection (or transmission) at its surface went directly into another photodiode P2 (or P3) after passing through a diaphragm, a converging lens and bandpass filter. The main purpose of the diagrams was to separate the reflection (or transmission) beams from the front and the back of the sample (only the first reflected beam is of interest).

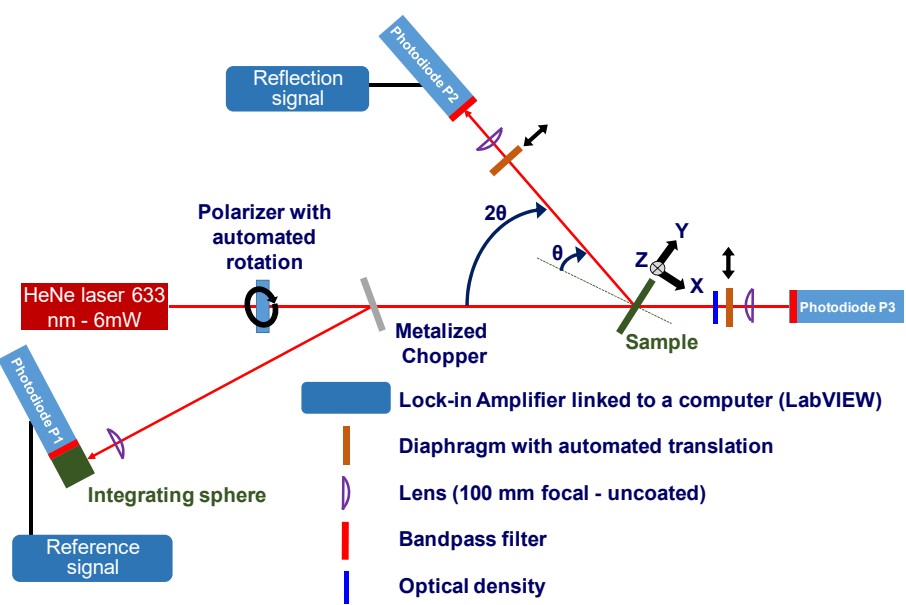

**Figure 3.** Scheme of the experimental ellipsometer.

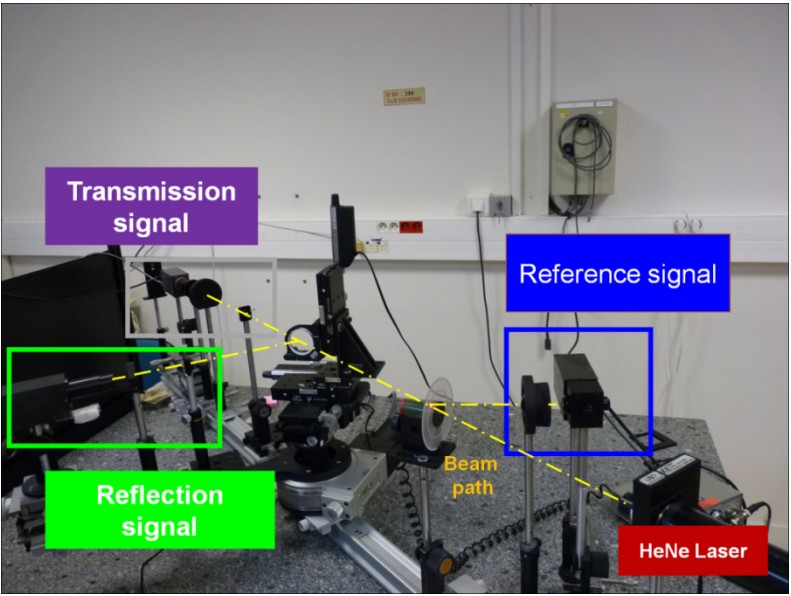

**Figure 4.** Picture of the experimental ellipsometer.

The photodiodes were equipped with bandpass filters to reject ambient light. These had a center wavelength of 633 nm. When a measurement was made in rotation at a given point, the reflected and transmitted beams were slightly shifted, even with the best bench setup. To avoid affecting the accuracy of the measurement, converging lenses ($f' = 100$ mm) were used to focus the reflected (transmitted) beam onto the same point on the photodiode.

The sample was placed in a holder having five independent degrees of freedom ($\theta_s$, $\varphi_s$, $x_r$, y and z). $\theta_s$ and $\varphi_s$ were used to perform the autocollimation of the sample. The motorized stage $x_r$ improved the accuracy of the sample surface on the rotary axis. The system enabled single measurements at a given position or mappings thanks to both y and z motorized stages. This holder was concentric to both rotation stages. It allowed the first one to orient the sample, and the second one to move a measuring arm around the sample. An XPS controller from MKS Company (Andover, MA, USA) drove all the motorized stages.

A LabView program piloted this ellipsometer and all the data from signal measurements were stored in a text file.

## 2.4. Ormosil Preparation by Sol–Gel

A great range of ormosils of the TEOS/PDMS system can be prepared by varying the PDMS concentration or the length of PDMS chains. The synthesis description and chosen formula are based on Compoint's works [15].

The synthesis of the PDMS-based ormosil solution was carried out by a sol–gel process and is similar to the synthesis of bulk silica-PDMS ormosils described by Compoint (2015) [15], by Maczenkie et al. (1996) [28] and Huang et al. (1986) [47], giving, in the end, a sol that can be deposited as a thin film by spin or dip coating Tetraethyl orthosilicate (TEOS) was chosen as a silica precursor for the inorganic component. A commercial PDMS with hydroxyl end groups (Si–OH) was selected as the precursor for the organic component. We have chosen to study ormosils made with a PDMS average molecular weight of 550 g·mol$^{-1}$, which corresponds to seven repetitions ($n = 7$) of the monomer in the polymer HO–[Si(CH$_3$)$_2$–O]$_n$–Si(CH$_3$)$_2$–OH. Absolute ethanol (99.9%) and tetrahydrofuran THF (99%) were used as solvents, while hydrochloric acid HCl was used as the catalyst. The aforementioned chemicals were purchased from Sigma-Aldrich Company (Saint-Louis, MO, USA). Various amounts of PDMS were added to the ongoing sol–gel synthesis.

A starting solution of TEOS, PDMS, and THF was first prepared under stirring while a mix of water, acid, and ethanol was introduced. In this study, the PDMS ratio introduced is expressed in the same way as in Mackenzie et al. (1996) work [28] in mass percent (wt.%) relative to the sum of TEOS and PDMS masses introduced (=PDMS/(TEOS + PDMS)). PDMS ratio varies up to 50 wt.%. The solution was heated to 80 °C under reflux for 6 h under mechanical (200 rpm) or magnetic (350 rpm) agitation according to a protocol inspired by Compoint and Mackenzie et al. syntheses [15,28].

The reaction, which leads to the formation of the silica network, is a TEOS hydrolysis in which the ethyl groups of the TEOS are switched with hydrogen to form Si–OH groups. Then, a condensation of the TEOS precursors occurs between the hydrolyzed species under acid catalysis to form the silica network. The PDMS chains react with the Si–OH (silanol) groups of the hydrolyzed TEOS or with the silanol groups on the surface of the formed silica network. Generally, the PDMS chains graft on the silanol group is made available by the synthesis conditions. The two reactions (formation of the silica network and grafting of PDMS chains on silanol groups) occur simultaneously in this co-polymerization. In the end, the reaction between the silica and the PDMS stays incomplete. Some PDMS chains remained free in the network, but they could be integrated into the structure by weaker, reversible hydrogen bonds.

The used PDMS had according to Sigma-Aldrich Company, the supplier, an average molecular weight of 550 g·mol$^{-1}$. Between the different packages purchased the weight could vary because the packages were composed of several lengths $n$ of molecular chains. Each molecular chain length corresponds to $n$ repetitions of the monomer in the polymer HO–[Si(CH$_3$)$_2$–O]$_n$–Si(CH$_3$)$_2$–OH, with $n = 7$ if the molecular weight is 550 g·mol$^{-1}$. To be sure of our used PDMS, Compoint measured a package by the Steric exclusion chromatography technique. These analyses [15] indicated the molecular weight, and the found molecular weight of the package was 686 g·mol$^{-1}$ or $n = 9$. After measurements with a PDMS having a molecular weight of 2500 g·mol$^{-1}$ a gap of 2 monomers ($n = 37$ from Steric exclusion analyses and $n = 35$ from Sigma-Aldrich Company) was also found by Compoint [15]. This showed that it is preferable to make a preliminary measurement of the molecular weight of the PDMS rather than using the approximate values provided by the supplier.

## 2.5. Sample to Highlight the Sol–Gel Layer Viscosity

To confirm the viscosity phenomenon responsible for self-healing observed in Figure 1a 40 wt.% PDMS-based ormosil of 2150 nm thickness was spin-cast onto a fused silica substrate of 5 mm thickness and 50 mm diameter. Only part of the surface was coated to increase the phenomenon at the interface. Figure 5 describes the sample studied and the location of the mapping measurement on it. The mappings were carried out at different times between day 0 and day 38 for two different angles of incidence (45° and 50°). These

two angles of incidence were used because Zhao et al. (2011) showed that the inaccuracy of ellipsometry increases with the increase in incidence [42].

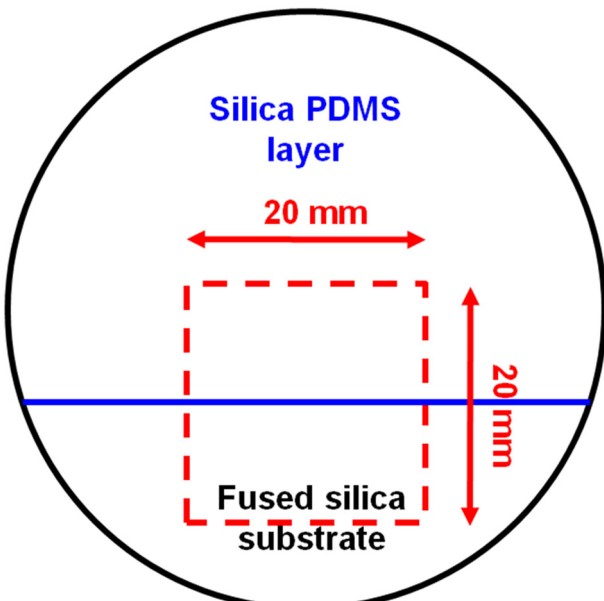

**Figure 5.** Scheme of the studied sample with the measurement area of $20 \times 20$ mm$^2$ approximately in the center of the film boundary sample.

The sample was measured in a vertical position to reinforce this phenomenon using gravity.

## 3. Results

### 3.1. Development of Our Ellipsometer

The chopper was metalized, so the reflection on its surface could be used as a reference signal to consider temporal laser fluctuations. The reflected signal is recorded over time and is drawn in Figure 6.

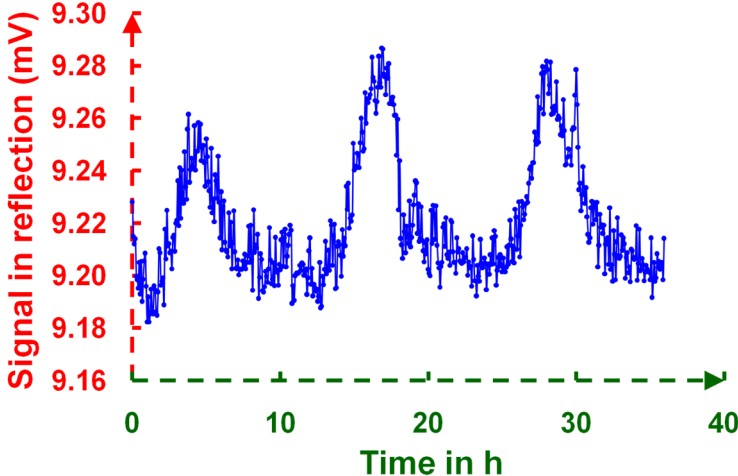

**Figure 6.** Laser fluctuations over a long time period (35 h).

Figure 6 clearly showed the output laser power variations. Its fluctuation range was 104 µV, corresponding to 0.256% relative fluctuation. As the chopper reflection was very diffuse, due to the metalization and flatness of the chopper disk, the beam has been focused with a convergent aspheric lens (50 mm diameter) on a photodiode through an integrating

sphere (LABSPHERE ISO-SFAIG from Labspshere Inc. (Sutton, NH, USA)). After numerous reflections in the integrating sphere, the beam went out through the output port and was analyzed by a photodiode to give the reference signal.

Figure 7 illustrates the measurements on the reflected and reference arms. Both signals vary over time but are well correlated. It was thus possible to correct the temporal fluctuations, as indicated in Figure 8, where the ratios between reflected and reference signals were calculated. The root mean square of these ratios was around 0.054%, five times smaller than for the reflected signals without correction.

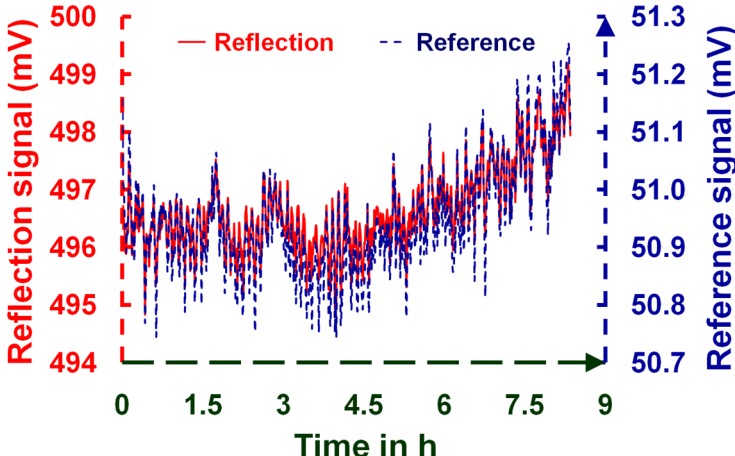

**Figure 7.** Laser fluctuations measured on both channels over 9 h.

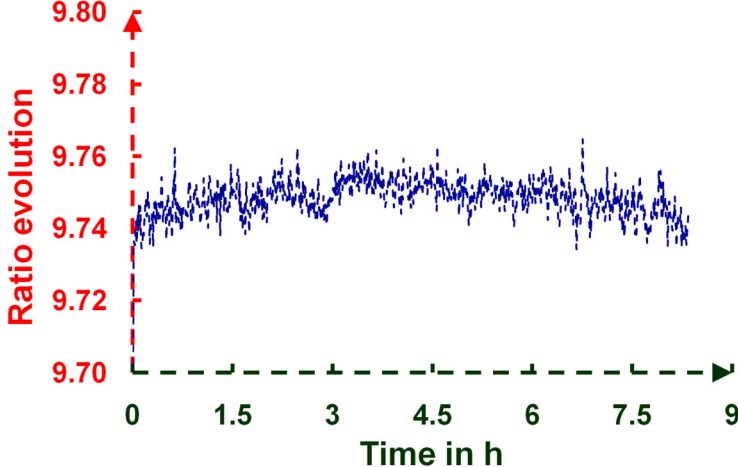

**Figure 8.** Signal fluctuations with a ratio between the reflection and the reference signals over 9 h.

### 3.2. Determination of $K_\lambda$

We also measured the noises from the photodiodes over time. These measurements are presented in Figure 9.

To determine $K_\lambda$($\lambda$ = 633 nm) precisely, fused silica and BK7 substrates were measured as a function of the angle of incidence in p-polarization. Their experimental curves were adjusted with theoretical curves obtained by simulation. The results are drawn in Figure 10.

Table 1 summarizes the calibration results of fused silica and BK7. The refractive indexes were computed for both substrates from Brewster's angle method described above. These results are compared to refractive index values from the reference at 633 nm [48,49] and with values obtained from the RC2 ellipsometer.

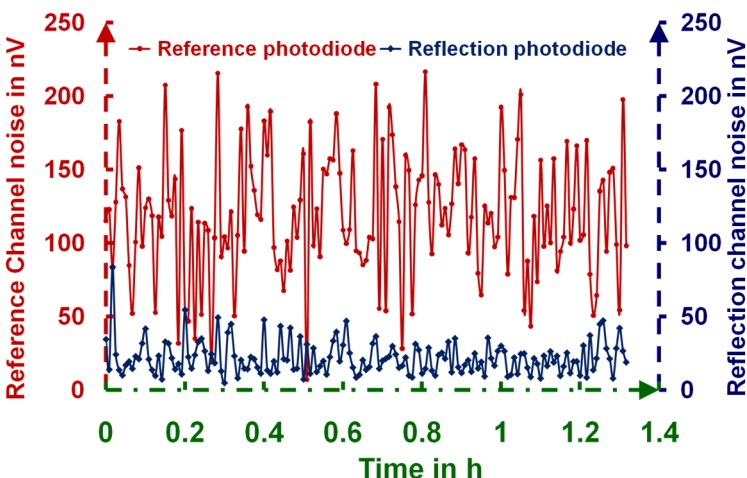

**Figure 9.** Photodiode noise with daylight and interference filter.

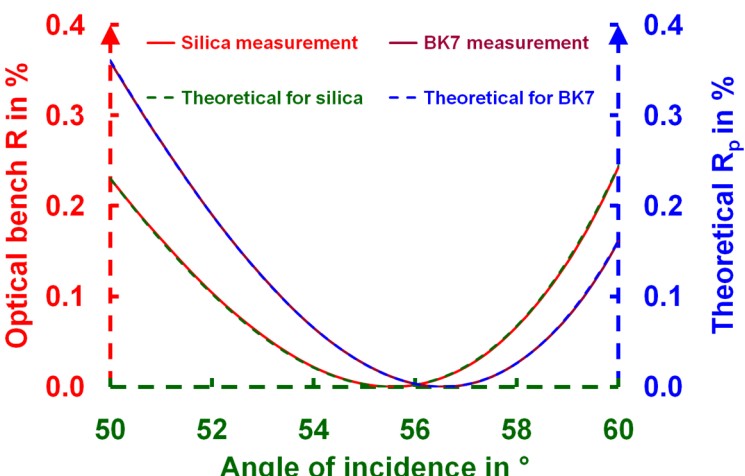

**Figure 10.** Measurements and simulations of a silica substrate and a BK7 substrate in p-polarization.

**Table 1.** Summary of different results from Figure 8 and the calculated value of the coefficient $K_\lambda$.

| Material | Silica | | BK7 | |
|---|---|---|---|---|
| Brewster's Angle<br>$K_\lambda$ (633 nm) | 55.53<br>0.112767 | | 56.54<br>0.112662 | |
| | n (633 nm) | % difference from reference | n (633 nm) | % difference from reference |
| n (633 nm) reference | 1.4571 from Palik [48] | 0% | 1.5150 from supplier [49] | 0% |
| n (633 nm) RC2 | 1.4576 | 0.03% | 1.5149 | 0.01% |
| n (633 nm) homemade | 1.4566 | 0.03% | 1.5131 | 0.12% |

*3.3. Thin Film Thickness Determination*

After determining the refractive index of the layer with Brewster's angle, the theoretical value of the thin film reflectance $R_p(\theta$ at 633 nm) [38,39] was calculated as a function of the layer thickness for a fixed angle of incidence (Figure 11 at $\theta$ = 45°). The theoretical thickness, corresponding to the experimental $R_p$, was thus determined. However, the reflectance is a surjective function, meaning, as illustrated in Figure 11, that the same reflectance $R_p$ value can be associated with several different thicknesses.

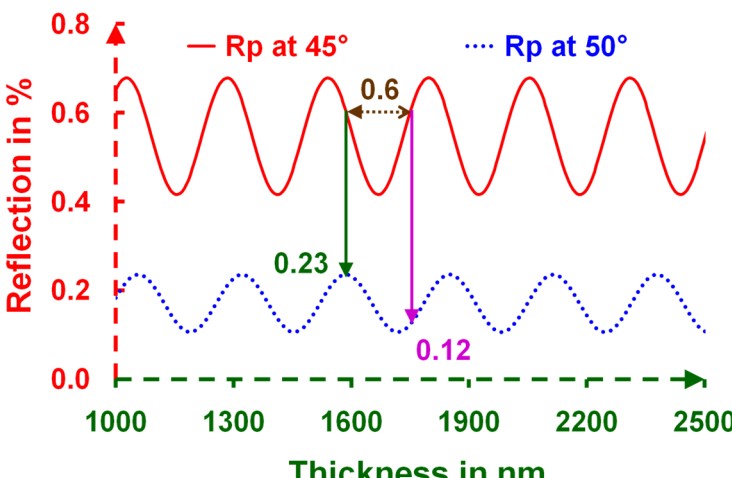

**Figure 11.** Reflection in p-polarization for two angles of incidence (45° and 50°) at 633 nm of a layer with a refractive index of 1.421 deposited on a substrate with a refractive index of 1.459.

Thus, the local thicknesses were determined with both measurements at two different angles of incidence. Figure 12 gives an example of a comparison between the $R_p$ determined experimentally and the one calculated theoretically (t = 1935 nm) for a given thickness. The green and blue curves were computed for thicknesses other than the black ones, and only the black curve fit the experimental $R_p$.

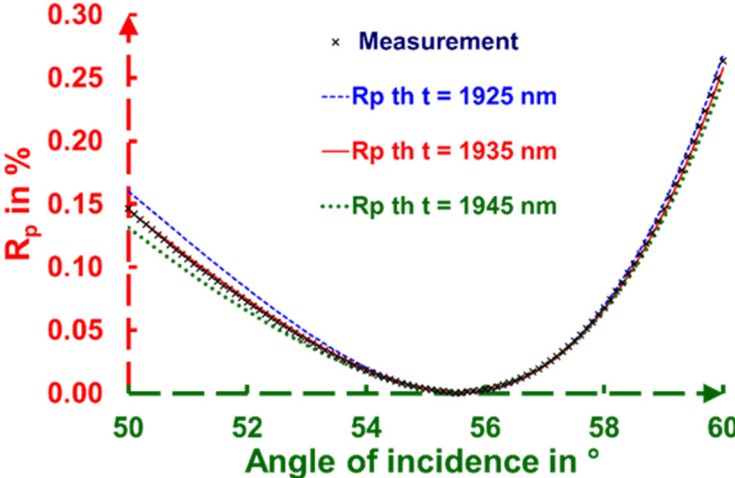

**Figure 12.** Comparison between an angular measurement and its fit ("th" stands for theoretical) corresponding to a local thickness t = 1935 nm of the PDMS-based ormosil layer. We had two other examples of $R_p$ calculation for a thickness t = 1945 nm and t = 1925 nm to indicate the measurement accuracy of the setup (<10 nm).

*3.4. Example of a Heterogeneity Thickness of PDMS-Based Ormosil Layer Compared to Silica Substrate*

By repeating this process to an equally spaced set of points across the samples' surface, film thickness homogeneity maps were obtained. Figure 13 gives an example of a typical "reflectance" map and its associated thickness map made with the precedent sample (Figure 12). The thickness heterogeneity of this measured layer was visible. The measurement zone was delimited by a 20 mm square in the middle of a 50 mm-diameter sample. The measurement was performed at 45° and 50° in s-polarization. The spatial resolution was 2 points·mm$^{-1}$.

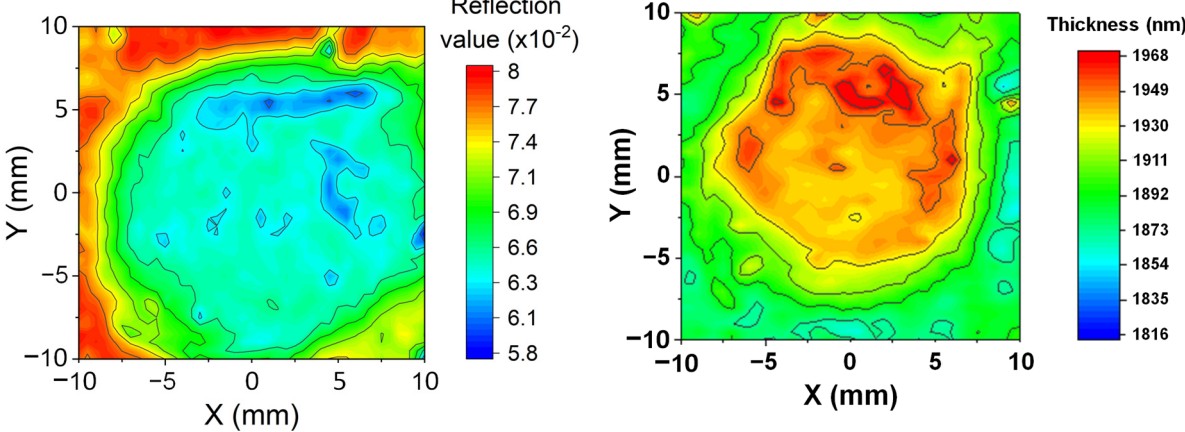

(**a**) Reflectance map obtained at 45° in s-polarization.　　(**b**) Thickness map of a PDMS-based ormosil film.

**Figure 13.** Homogeneity of reflection (**a**) and thickness (**b**) of a 40 wt.% PDMS-based ormosil layer.

Figure 14 displays the map of

$$m = 100 \cdot \frac{R_p - \text{Average}}{\text{Average}} \tag{10}$$

drawn with identical dimensions to Figure 13 on a polished silica substrate at 50° but for p-polarization.

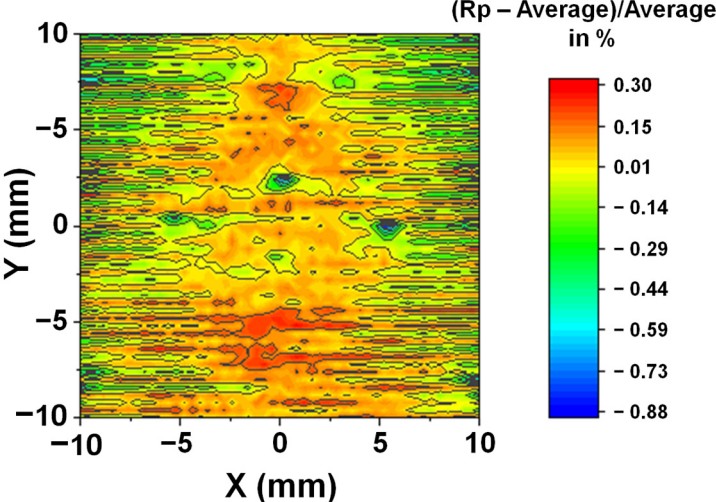

**Figure 14.** Map of m, the relative homogeneity from the reflection of silica sample at 50° ($R_p$ = 0.231%).

Table 2 gives the percent of the surface measured whose relative variation m is less than |x| (with x = 0.1, 0.2, 0.3, and 0.5%).

**Table 2.** Relative variation in reflection of the silica sample measurement in p-polarization.

| Gap | −0.1% < m < 0.1% | −0.2% < m < 0.2% | −0.3% < m < 0.3% | −0.5% < m < 0.5% |
|---|---|---|---|---|
| % on Measured Surface Silica Inside the Gap | 76.6 | 88.1 | 94.5 | 99.3 |

### 3.5. Viscoelasticity Effect Highlighted by Our Ellipsometer

The refractive index of this layer made specifically was determined with our ellipsometer, as explained previously, and found to be 1.42. Several mappings are carried out in p-polarization for 38 days for two incident angles (45 and 50°) and translated in thickness. Figure 15 depicts the relative thickness gaps between day 0 and day 38 (Figure 15a) and a zoom of the map, here expressed in thickness (Figure 15b). A more detailed study was performed along the Y axis at a fixed X (pink line drawn in Figure 15a). With theoretical calculations, Figure 17 shows reflectivity variations with thickness variations for the 40 wt.% PDMS-based ormosil, with a thick layer of 2150 nm.

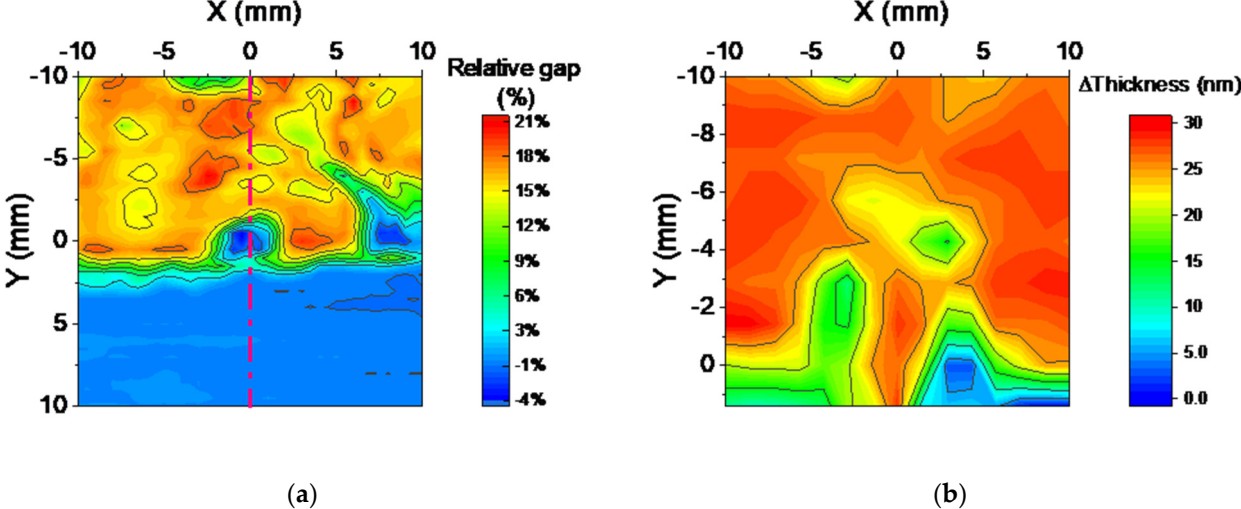

(**a**)          (**b**)

**Figure 15.** Thickness variations due to the viscosity of the ormosil layer. (**a**) Relative thickness gaps between day 0 and day 38. These gaps are due to the viscosity of the 40 wt.% PDMS-based ormosil thin film. The pink line represents the axis of the study depicted in Figure 16. (**b**) Thickness map of the relative thickness gaps between day 0 and day 38.

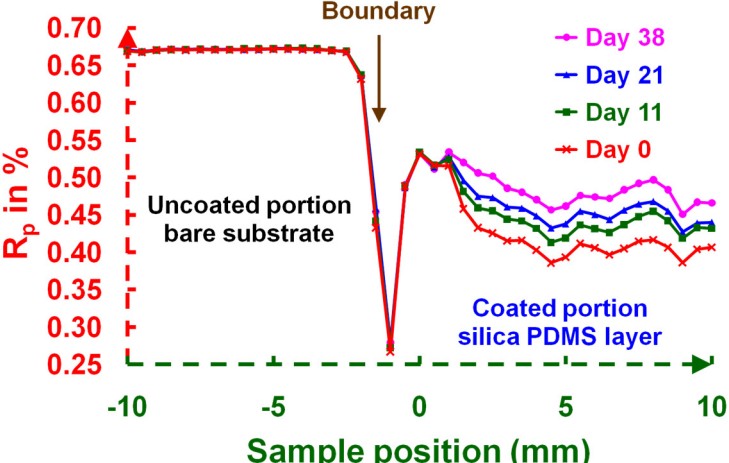

**Figure 16.** $R_p$ coefficient as a function of the direction Y, as indicated by the pink line (X = 0 mm) drawn in Figure 15a.

In Figure 16, the thickness evolutions over time along the pink line X = 0 mm in Figure 15a are drawn.

Table 3 summarizes the evolution of average deviation and associated thickness decreases during the 38 days. The averages are made between 2 and 10 mm.

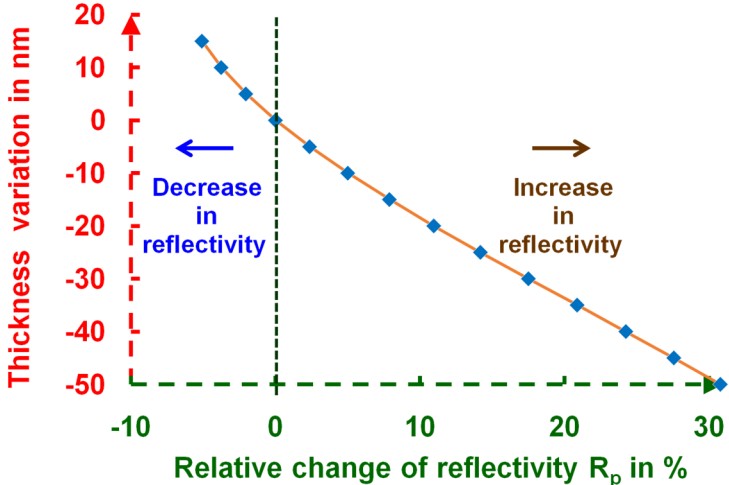

**Figure 17.** Theoretical reflectance variation as a function of thickness. An increase in $R_p$ indicates a thickness decrease compared to the nominal one.

**Table 3.** Average differences for different days compared to the initial map (2 mm < Y < 10 mm) and associated thickness decreases.

| Time | Average Deviation from the Initial Map 0 | Associated Thickness Decrease |
|---|---|---|
| Day 0 | 0% | 0 nm |
| Day 11 | 7.16% | 13.3 nm |
| Day 21 | 10.96% | 19.7 nm |
| Day 38 | 17.23% | 29.8 nm |

*3.6. Confirmation of the Layer Evolution with Our Woollam Ellipsometer*

Two mappings of a 40 wt.% PDMS-based ormosil film were performed one year apart from each other. Figure 18 depicts these maps made in 2021 and 2022. The colors chosen in Figure 18 are identical in the two maps.

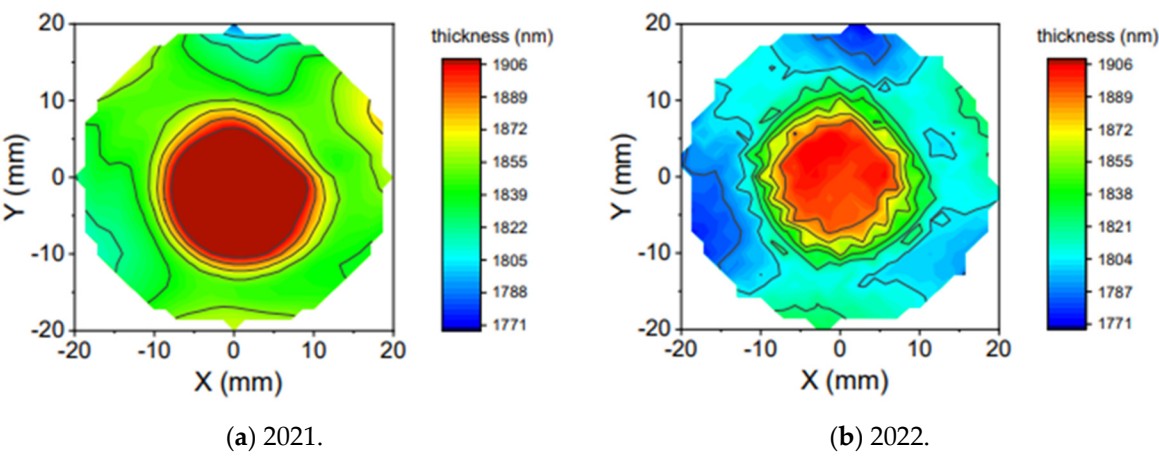

(**a**) 2021.                                                          (**b**) 2022.

**Figure 18.** Maps of the same thin film (40 wt.% PDMS-based ormosil) at two different dates: (**a**) in 2021 and (**b**) in 2022. The sample was stored in a horizontal position.

## 4. Discussion

*4.1. Development of Our Ellipsometer*

To improve the setup, two diaphragms (one on the reflection path, and another on the transmission path) were mainly used to separate the reflected beams coming from the front and the back face of the sample (the first reflected beam being the only one of interest). The

positions and the diameters of both diaphragms depended on the orientation of the sample because the reflected beam shift,

$$\delta = \frac{t \cdot \sin 2\theta}{\sqrt{n^2 - \sin^2 \theta}} \tag{11}$$

Ref. [42], is linked to the angle of incidence θ, the refractive index *n*, and the sample thickness t. 5 mm transparent thick samples were used to increase this shift. A lens and an interference filter (10LF10-633-B from Newport Company, CWL of 633 nm, and FWHM of 10 ± 2 nm) were added on both paths to suppress the little beam shift coming from a tunable defect.

Figure 6 indicates high fluctuations of the laser source on the measurement arm. The signal coming from the reflection of the laser source on the metalized chopper was perfectly correlated with the measurement arm (Figure 7). It can be used as a reference signal. The fluctuations were reduced by the ratio of these two signals by a factor of five (Figure 8).

### 4.2. Determination of $K_\lambda$

The measured reflection R(θ) is given by the relationship between reflectance and the other signals [31,32]. It links from the voltage values $U_i$ to reflection coefficients R(θ).

$$R(\theta) = K_\lambda \frac{U_{reflection}(\theta) - U_{reflection\ background}}{U_{reference} - U_{reference\ background}} \tag{12}$$

where $K_\lambda$ is the calibration constant, $U_{reflection}(\theta)$ is the signal from the reflected path for an incidence angle θ, and $U_{reference}$ is the signal from the reference path. $U_{reflection\ background}$ and $U_{reference\ background}$ are signal noises from the reflected and reference parts, respectively.

The noises $U_{reflection\ background}$ and $U_{reference\ background}$ from the photodiodes have been evaluated, and they are drawn in Figure 9. They were completely random, with an order of magnitude of $10^{-7}$ V. The measurements were performed during 1.35 h. As shown, in Figure 9, photodiode noises were negligible compared to fluctuations from measurements of thin films. Therefore, they were neglected afterwards and R(θ) became.

$$R(\theta) \approx K_\lambda \frac{U_{reflection}(\theta)}{U_{reference}} \tag{13}$$

To determine $K_\lambda(\lambda = 633 \text{ nm})$ precisely, fused silica [48] and BK7 [49] substrates were used. Their refractive index, n, was retrieved from Brewster's angle ($\theta_B$) measurement (n = tan $\theta_B$). The reflection was recorded as a function of the angle of incidence, and the corresponding curve was fit with a polynomial function. Brewster's angle was determined by finding the value of the incident angle at which this reflection was minimal. Knowing n, the theoretical reflection $R_p(633 \text{ nm})$ was calculated from homemade software based on Abelès formalism [38] as a function of the angle of incidence. A VBA (Visual Basic for Applications) macro was used to fit the measurement to the theory and determined $K_\lambda$. Figure 10 illustrates the excellent correlation between measurements made on fused silica and BK7 substrates with the corresponding theoretical simulations.

From Figure 10 and Table 1, we can conclude that refractive indexes measured on the experimental ellipsometer for BK7 and fused silica are close to the reference values (from Palik and suppliers) and the measured value given by the RC2 ellipsometer, which validates the setup. So, the coefficient $K_\lambda$ from the relationship (13) is determined. Hereafter, the value taken for $K_\lambda$ is $K_\lambda(633 \text{ nm}) = 0.112715$ (mean value obtained). Now, we can obtain the reflection coefficient of a sample from any angle of incidence and polarization.

### 4.3. Thin Film Thickness Determination

Figure 11 indicates that in some cases it may have several possible thicknesses. To avoid this problem, each experimental measurement was done at two different angles of

incidence (45° and 50°) to use the reflectance value associated with the other angle when a conflict appeared with the first value of reflectance at the first angle of incidence.

Figure 12 also shows how sensitive these measures are. Two reflection curves were computed around the correct value of thickness t = 1935 nm, one at t = 1925 nm, and another at t = 1945 nm. In the right part of Figure 12, a reflection gap is clearly distinguished between both curves and the curve corresponding to the correct thickness. A VBA macro was used to automatize the calculations and to work out the final thickness corresponding to the average thickness obtained for all the angles of incidence. The thickness sensitivity of our ellipsometer was well above 10 nm.

### 4.4. Example of A Heterogeneity Thickness of PDMS-Based Ormosil Layer Compared to Silica Substrate

In p-polarization, and at a 50° incident angle, the theoretical reflectance factor at the wavelength 633 nm is 0.231% (see Figure 10). In these experimental conditions, we have mapped a polished silica sample and displayed the map of m in Figure 14. We cannot overlook that a measured silica substrate must have local defects. They are responsible for major local differences that we observed in Figure 14. Nevertheless, they are included in the given results in Figure 14 and Table 2. They highlight the quality of our measurements. According to these results, we can tell that the PDMS-based ormosil layer is highly heterogeneous, as shown in Figure 13. The thickness gap is 151 nm or 8% of the average thickness of a measured surface of 20 × 20 mm$^2$. This heterogeneity comes from the viscosity of the sol-gel solution and solvent evaporation. It increases when increasing the surface size. As this layer is planned to be deposited on LMJ optical components whose sizes were 400 × 400 mm$^2$, the viscosity, and the solvent evaporation would lead to heterogeneity variations not meeting the LMJ requirements.

### 4.5. Viscoelasticity Effect Highlighted by Our Ellipsometer

For each mapping at each angle of incidence (45° and 50°), the thickness reflection gap was calculated by comparing the value of the latest map to the first one made on the first day 0. As time passed, the measurement gap widened, as shown in Figure 15. A relative gap of up to 21% (Figure 15a) was measured after 38 days. As expected, the experimental setup was sensitive to thickness variations and way above signal deviation due to measurement repetition or noise. Thickness changes occurred on the entire coating and not specifically at the interface between the layer and the substrate. The relaxation mechanisms derived from the viscoelastic properties of the coating had a uniform effect on the entire surface of the coating and not only on the area near the interface (coated/uncoated). After 38 days, the thickness changes reached 30 nm over the treated area we measured (Figure 15b).

The relative change of reflectivity was given by:

$$100 \cdot \frac{R_P(x) - R_P(\text{reference})}{R_P(\text{reference})} \tag{14}$$

For a layer near 2150 nm at 45°, the relative change of reflectivity is almost linear with thickness variation in this thickness range. For each increase in the relative gap of reflection, layer thickness decreases as indicated in Figure 17. However, as seen later on, the thickness variation was not only visible at the interface, but also on the entire layer surface (Figure 16).

In this case, for the specific lines studied (Figure 16), an average deviation from the initial map was 17.23% at day 38, with an associated thickness decrease of around 30 nm. This deviation from the nominal thickness went crescendo as $R_P$ increased over time ($R_P(\text{Day } 38) > R_P(\text{Day } 21) > R_P(\text{Day } 11) > R_P(\text{Day } 0)$) as indicated in Table 3. The hole in the reflection at the boundary (Figure 16) was due to the presence of a local defect.

In fact, 38 days were not enough time to see all the thickness variations, which were accentuated by the vertical positioning of the sample. Nevertheless, these thickness variations will be a problem in the design of a future antireflective coating on this elastic

layer if we do not want to distort too much the wavefront of the power laser at the output of the optical component. This effect is shown in Figure 2.

This elastic layer must also be covered by a thin layer to have an antireflective component resisting laser flux. It would be interesting to study the effect of viscosity on the entire stack in the future.

*4.6. Confirmation of the Layer Evolution with Our Woollam Ellipsometer*

A commercial spectroscopic ellipsometer, received in the meantime, was used to perform thickness mapping at 633 nm for an angle of incidence of 50°. Two mappings of a 40 wt.% PDMS-based ormosil film were performed one year apart from each other (Figure 18). The thickness maps were acquired on a 40 mm diameter circle with a $29 \times 29$ point pattern (1.38 mm spacing between each point). The sample was stored horizontally in the same position at room temperature and constant humidity for one year. The colors of Figure 18 were chosen to highlight thickness changes between 2021 and 2022. Figure 18 shows well a decrease in layer thickness over time due to its viscoelasticity, which confirms that ormosil layers can have thickness variation over time. For this layer, the thickness variation $\Delta t$ over a year is $\Delta t = 40$ nm, which is in the same range as our previous measurement on the experimental ellipsometer.

**5. Conclusions**

PDMS-based ormosil films were developed for shockwave mitigation to protect the optical components of the LMJ. The thickness of these layers was a few µm, and the PDMS-based ormosil films show viscoelastic properties. This µm thickness induces quicker phase variations than thinner antireflective coating in the context of the same relative thickness heterogeneity. An experimental ellipsometer has successfully been developed to control thickness heterogeneity and to measure its evolution over time.

The experimental ellipsometer was developed to perform measurements in a vertical position and validated with BK7 and fused silica substrates. It measured the refractive index and thickness of thin films with a sensitivity under 10 nm.

Mappings on 2 µm thick ormosil thin films showed a 150 nm thickness surface heterogeneity due to the solution's spin coating technique and viscosity.

Studies of these thickness variations over time were led on samples held vertically and confirmed the viscoelasticity of these PDMS-based ormosil films. Thickness variations were up to 30 nm after 38 days.

Another study performed on a commercial ellipsometer showed that this viscoelasticity also leads to a variation in thickness when the samples are held horizontally.

Experiments are currently carried out using a picosecond acoustic technique to determine the viscosity values of these thin films.

According to these results, further research will have to be conducted on the stability of these thickness variations of viscoelastic layers to respect LMJ specifications [50].

**Author Contributions:** Conceptualization, H.P. and A.G.; methodology O.G. and A.G.; software, O.G.; validation O.G., A.G. and H.P.; formal analysis, M.L.; investigation, T.B., A.M. and M.L.; resources, H.P.; data curation, H.P.; writing—original draft preparation, M.L., A.G. and O.G.; writing—review and editing, A.G. and O.G.; visualization, O.G., A.G. and H.P.; supervision, H.P.; project administration, H.P.; funding acquisition, H.P. All authors have read and agreed to the published version of the manuscript.

**Funding:** This research received no external funding.

**Institutional Review Board Statement:** Not applicable.

**Informed Consent Statement:** Not applicable.

**Data Availability Statement:** Data underlying the results presented in this paper are not publicly available but may be obtained from the authors upon reasonable request.

**Conflicts of Interest:** The authors declare no conflict of interest.

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
