# Peer review of "Transparent Films Thickness Mapping Highlighting the Viscosity Effect of Elastic Layers Made by Sol–Gel Process with an In-House Ellipsometer"

_coatings, doi:10.3390/coatings13030633_

Round 1
Reviewer 1 Report
Please replace the french words in Figure 1 with English one !
Please define all acronyms before their first appearance in text i.e CEA/MAD, PDMS and so on
Please avoid such big block citations “[4, 7, 9-16]”
“are not stable over time because of their viscosity, they evolve.” For this do you propose any future solution for controlling it ? Or provide a prediction how they evolve ?
the phase shift is about 4.6° at 358 nm. – your graph do not show it !
“ellipsometer was quickly developed” please reformulate this as a device cannot be designed quickly !
The description of Figure 3 and 4 is quite vague a better description is required for nonprofessional and peoples which do not work in the field
“It is an average value and does not correspond to the real one as it fluctuates between” this statement can be used in lab discussion but not in a research paper as the other researchers want to trust in your methodology
“The value found is 686 g.mol−1 instead of the commercial value of 550 g.mol−1,” it seems very big difference and your explanation is quite vague about the difference !
Most of references are out of date- some recent references are required too!
Author Response
"Please see the attachment."

Author Response
"Please see the attachment."

Reviewer 3 Report
The motivation for the paper should be expanded upon in a redraft. The authors designed an ellipsometer that allows the sample to be measured in a vertical orientation as compared to a Woolam ellipsometer which requires the samples to be measured in a horizontal orientation. The paper compares the temporal changes in thickness of an ormosil material that could serve as a shock-absorbing layer between a silica substrate and an anti-reflective layer used in optics for a high power laser. The manuscript shows that the ormosil changes over time due its viscoelastic properties whether the measurements are made with the sample vertical (Fig. 15) or horizontal (Fig. 18). Additional justification is needed from the authors for this new method since it provides a result that can be anticipated using known methods.
In addition, there are also several items that should be cleaned up in the manuscript is re-submitted. First. the paper talks about reflectance values value and no polarization (e.g., Rp) is assigned to experimental values in the text and figures (e.g., Fig. 10, Fig. 11). In contrast, theoretical calculations are generally performed on the reflectance of p-polarized light (Rp). The text should make clear the polarization of the light in all experimental readings. A second items to correct is that measurements were taken at incidence angles of 45 and 50 degrees. However, in many instances, the text drops the term "incidence angle" and just gives a (presumably incidence) angle (e.g., Fig. 13. A third item that needs to be corrected in the molecular weight of PDMS. The paper gives two conflicting values, 500 and 686. The latter one was determined experimentally whereas the former one was from the PDMS supplier. The authors should pick a value. A fourth area for additional improvement is a better description of how the sample area was created (see Figure 5 and Figure 17). It is not clear from the manuscript how the authors coated such a precise area as suggested by Figure 5.
Author Response
"Please see the attachment."

Reviewer 4 Report
You will find a review of the article in the attached file.

Author Response
"Please see the attachment."
Reviewer 5 Report
I can recommend the publication of the manuscript after a minor revision.
Write in alphabetical order the keywords.
Line 24, 26, 29, 185: “... [4-9]., .. [4, 7, 9-16].”, “[17-20]”, ...[49-54]”... and so on.
You will likely need to re-write your citation sentences, rather than simply replacing the numbers with Authors’ names. This is due to the fact that in order to give readers the maximum appreciation of how your work builds on previous results, each one of the cited sources should be discussed individually and explicitly to demonstrate their significance to your study. We ask that you use the authors' surnames as the subject of a verb, and then state in one or two sentences what they claim, what evidence they provide to support their claim, and how you evaluate their work. We also, therefore, ask that you avoid citing more than one reference in one sentence. This will give you a chance to discuss each reference separately.
What we are asking for is something like: “Smith (2011) describes the development of a finite element model of hot forging and claims excellent agreement between the model and experiments. However, he tests only one operating condition, tunes his model by modifying the friction coefficient, and compares only the total tool force. A much more detailed comparison would be required to evaluate the precise conditions under which finite element modeling is truly accurate."
Line 76: Fig. 1c (change “jours“ with “days”).
Insert references for all mathematical formulas.
Lines: 199, 404-406, 414-415, 430-433, insert more explanations and details.
Lines 471-472: Delete the sentences “For research articles with several authors, a short paragraph specifying their individual contributions must be provided. The following statements should be used”.
Lines 485-486. Delete the sentence “The funders had no role in the design of the study; in the collection, analyses, or interpretation of data; in the writing of the manuscript; or in the decision to publish the results.” because you have no funders.
Minor mistakes in references (e.g. ref. [4], [10], [42], and so on).
Incomplete information in ref. [56].
Even though the work is relevant to the journal's scope, i.e., Coatings, I do not find even a single article published in the journal in the list of references.
Could you insert more up to dated references?
I will recommend the present scientific manuscript for further publication once all the suggestions mentioned above are properly fixed.
Author Response
"Please see the attachment."

Round 2
Reviewer 1 Report
.
Reviewer 2 Report
I thank authors for revising the paper according to reviewer suggestions.